# ABSTRACTING INFLUENCE PATHS FOR EXPLAINING (CONTEXTUALIZATION OF) BERT MODELS

## ABSTRACT

While *"attention is all you need"* may be proving true, we do not yet know *why*: attention-based transformer models such as BERT are superior but how they contextualize information even for simple grammatical rules such as subject-verb number agreement (SVA) is uncertain. We introduce *multi-partite patterns*, abstractions of sets of paths through a neural network model. Patterns quantify and localize the effect of an input concept (e.g., a subject's number) on an output concept (e.g. corresponding verb's number) to paths passing through a sequence of model components, thus surfacing how BERT contextualizes information. We describe guided pattern refinement, an efficient search procedure for finding sufficient and sparse patterns representative of concept-critical paths. We discover that patterns generate succinct and meaningful explanations for BERT, highlighted by "copy" and "transfer" operations implemented by skip connections and attention heads, respectively. We also show how pattern visualizations help us understand how BERT contextualizes various grammatical concepts, such as SVA across clauses, and why it makes errors in some cases while succeeding in others.

## 1 INTRODUCTION

Recent advancements in NLP have been spurred by contextualized representations created in deep neural models such as BERT (Devlin et al., 2019). These contextualized representations, which are designed to be sensitive to the context in which they appear (Ethayarajh, 2019), are also shown to capture many grammatical concepts (Lin et al., 2019; Tenney et al., 2019a), including subject-verb agreements(SVA) and reflexive anaphora(RA) (Goldberg, 2019). However, the exact mechanism of *contextualization* in BERT, i.e., the process of developing contextualized representations from representations of individual input words in the sentence context, remains unclear. For example, in the sentence `the pilots that the architect likes is/are short`, choosing the correct the verb `is` over `are` to agree with the subject requires contextualizing the verb with plurality information of the subject. In this paper, we answer the central question: **How is contextualization realized in BERT for grammatical concepts such as SVA and RA?** Specifically, can we identify sub-components of BERT that are a) sufficient for representing those concepts but also b) sparse enough to legibly show how BERT contextualizes the concepts across layers and whether the contextualization follows correct grammatical rules?

Prior works on explaining contextualization in BERT rely on the analysis of layer representations and attention components. Representation analyses, either by training a probing classifier (Lin et al., 2019; Tenney et al., 2019a), or finding parse trees embedded in the representations (Hewitt & Manning, 2019; Reif et al., 2019), demonstrate that relevant linguistic concepts are *associated* with the activations of BERT components (i.e. subject's number associated with the activations of a certain head at a certain layer, or subject's representation closer to that of the verb's under certain transformations), but do not tell us how representations come about inside the model. Meanwhile, inspection of attention weights as indicators of the flow of information between BERT layers (Clark et al., 2019), requires subjective inference of relevant function (i.e. inference that a certain head may be involved because high attention weights between cells at the subject and cells at the verb), which are found to be problematic in other contexts (Brunner et al., 2020; Jain & Wallace, 2019). Analysis of attention further disregards the role of skip connections that do not involve attention at all. Neither approach allows us to track a concept as a causal chain from input to output or to distinguish

helpful from hindering representations or flows (hindering information such as contextualization of confounding inputs like unrelated nouns in a sentence lead to errors on SVA).

To answer the central question while overcoming these limitations, we introduce *multi-partite*[1] *patterns*, abstractions of sets of paths through a neural model (a graph). Patterns quantify and localize the *effect* of an input concept (e.g. a subject's number) on an output concept (e.g. corresponding verb's number) to *a collection of* paths passing through a sequence of model nodes and/or edges. We describe *guided pattern refinement*, a search procedure for finding patterns representative of concept-critical paths that let us selectively explore the importance of chosen aspects of a model (e.g. in BERT, we can refine patterns showing criticality of certain heads to paths also showing whether this is due to skip connections or due to attention). To demonstrate the contextualization process, we further extend the experimental framework to integrate impacts of multiple words towards a given concept (as opposed to impact of a single word, e.g. subject on SVA).

**Contributions**: 1) We describe *multi-partite patterns* for explaining the model-wide contextualization in neural models like BERT and *guided pattern refinement* (GPR) to discover influential patterns focusing on model elements of interest. 2) We visualize BERT's contextualization grammatical concepts including subject-verb agreements(SVA) and reflexive anaphora(RA), and qualitatively show how BERT encode these concepts using grammatically correct or incorrect cues. 3) We validate the sufficiency and sparsity of derived patterns with model compression and concentration metrics, respectively.

We begin with a summary of requisite techniques in Sec. 2. We describe the core elements of our methodology in Sec. 3 and exemplify them for understanding BERT in Sec. 4. We elaborate on related works in Sec. 5 and conclude in Sec. 6.

## 2 BACKGROUND

We introduce the basics of the BERT architecture (Fig. 1) and the learning task subject to our work. We then discuss existing explanation devices and how they motivate our methods that follow.

**BERT** In BERT, let $L$ be the number of Transformer encoder layers, $H$ be the hidden dimension of embeddings at each layer, and $A$ be the number of attention heads. The list of input word embeddings is $\mathbf{x} \stackrel{\text{def}}{=} [\mathbf{x}_1, \mathbf{x}_1, ..., \mathbf{x}_N], \mathbf{x}_i \in \mathbb{R}^d$. We denote the output of the $l$-th layer as $\mathbf{h}^l_{0:N-1}$. First layer inputs are $\mathbf{h}^0_{1:N} \stackrel{\text{def}}{=} \mathbf{x}_{1:N}$. We use $a^{l,i}_j$ to denote the $j$-th attention head from the $i$-th embedding at $l$-th layer and $s^l_j$ to denote the skip connection that is "copied" from the input embedding from the previous layer then combined with the attention output. Probability scores for candidates of [MASK] are denoted by $\mathbf{y}_i \stackrel{\text{def}}{=} \texttt{softmax}(W\mathbf{h}^L_i), W \in \mathbb{R}^{C \times H}$ where $C$ is the vocabulary size. We denote the index of [MASK] as $m$. The layered architecture is presented in Fig. 1(left) and a detailed view of the transformer layer in Fig. 1(right). For further details, refer to Vaswani et al. (2017) and Devlin et al. (2019).

We focus on Masked Language Modeling (MLM) used in BERT pretraining: predict a masked word represented by [MASK] in a context sentence. The MLM task has been used to evaluate whether BERT learns linguistic concepts such as SVA by measuring if it assigns a higher probability for the correct verb (e.g. are in Fig. 2) than the incorrect verb (e.g. is) at the [MASK] position (Goldberg, 2019).

**Distributional Influence.** To explain a DNN's behavior, *distributional Influence* attributes to each input a measure of impact on model output. Saliency (Baehrens et al., 2010), as an example, defines influence as the gradient of output w.r.t. the input. In a generalized framework of Leino et al. (2018), influence quantifies the impact of each input feature towards a concept (e.g. SVA) by instrumenting a model's inputs with a *distribution of interest* (DoI) and the output with a *quantity of interest* (QoI).

**Definition 1 (Distributional Influence)** *Given a model $f : \mathbb{R}^d \to \mathbb{R}^n$, an input $\mathbf{x}$, a DoI $\mathcal{D}(\mathbf{x})$, and a QoI $q : \mathbb{R}^n \to \mathbb{R}$, Distributional Influence $g_q(\mathbf{x})$ quantifies the impact of an input concept defined*

---

[1]*Multi-partite* because patterns abstract sets of paths in neural models viewed as multi-partite graphs.

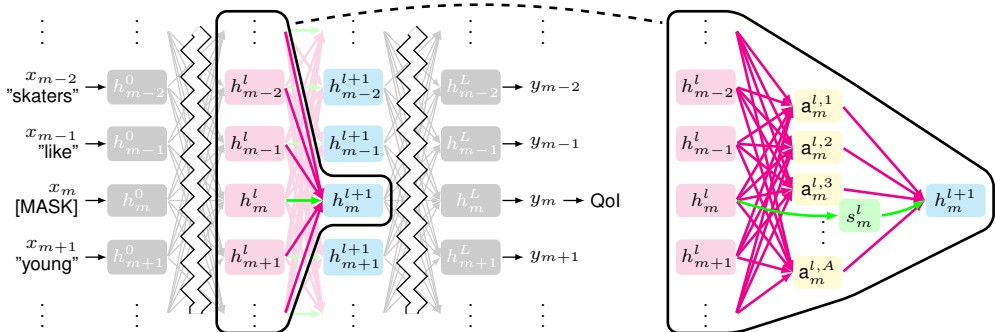

Figure 1: BERT Transformer architecture (left) and details of a transformer layer (right).

*by the DoI on the output concept defined by the QoI:*

$$g_q(\mathbf{x}) \stackrel{\text{def}}{=} \mathbb{E}_{\mathbf{z} \sim \mathcal{D}(\mathbf{x})} \frac{\partial q(f(\mathbf{z}))}{\partial \mathbf{z}}$$

Instantiations defining SVA and RA concepts in BERT models are found in Sec. 4. Examples of DoI include Gaussian distributions with mean $\mathbf{x}$ (Smilkov et al., 2017), or uniform distributions over a path $c = \{\mathbf{x} + \alpha(\mathbf{x} - \mathbf{x}_b), \alpha \in [0, 1]\}$ from a user-defined baseline input $\mathbf{x}_b$ to the target input $\mathbf{x}$ (Sundararajan et al., 2017). We use the latter in the rest of paper; we approximate the expectation in Def. 1 by sampling discrete points in the uniform distribution (Sundararajan et al., 2017) (see Appendix B.1 for an analysis of the accuracy of approximation).

**Explaining Contextualization with Influence Paths.** While it can highlight relevant inputs, *Distributional Influence* cannot show if or how they are contextualized internally to form higher-level concepts. *Influence Paths* (Lu et al., 2020) localizes an input influence measurement to paths in a neural model and thus can be used to show how the influence of the input representations flows internally through one internal representation to another. A computation graph $\mathcal{G} \stackrel{\text{def}}{=} (\mathcal{V}, \mathcal{F}, \mathcal{E})$ is a set of nodes, activation functions, and edges, respectively. In this paper, we assume the graph is directed, acyclic, and does not contain more than one edge per adjacent pair of nodes[2]. A path $p$ in $G$ is a sequence of graph-adjacent nodes $[p_1, p_2, \cdots, p_{-1}]$. We denote the Jacobian of the output of node $n_i$ w.r.t the output of connected (not necessarily directly) predecessor node $n_j$ evaluated at $\mathbf{x}$ as $\partial n_j(\mathbf{x})/\partial n_i(\mathbf{x})$ We write $\triangledown_{\mathbf{x}} p$ as the component of the Jacobian passing through path $p$ evaluated at input $\mathbf{x}$ as per chain rule: $\triangledown_{\mathbf{x}} p \stackrel{\text{def}}{=} \prod_{i=1}^{-1} \partial p_i(\mathbf{x})/\partial p_{i-1}(\mathbf{x})$.

**Definition 2 (Individual Path Influence)** *Given a path $p$ of a computation graph $\mathcal{G}$, the* individual path influence *for an input $\mathbf{x}$, or $\chi(\mathbf{x}, p)$ is:*

$$\chi(\mathbf{x}, p) \stackrel{\text{def}}{=} \mathbb{E}_{\mathbf{z} \sim \mathcal{D}(\mathbf{x})} [\triangledown_{\mathbf{z}} p]$$

Lu et al. (2020) uses individual path influence to decompose distributional influence to paths and explain internal LSTM behaviour under SVA via the most influential path $\arg\max_{p \in \mathcal{P}} \chi(\mathbf{x}, p)$ where $\mathcal{P}$ are all paths from input to a particular output (normally a QoI). The influence path approach relies on enumerating $\mathcal{P}$.

## 3 PATH ABSTRACTION

Directly applying individual influence paths of Lu et al. (2020) to transformer-based models like BERT has computational and conceptual problems. BERT is denser in terms of model connections: each node at each layer integrates information from *all* nodes of the prior layer (as opposed to the pair of short-term and long-term connections in LSTM). This results in an intractable number of influence paths to enumerate, even for processing the simplest of BERT variants.

---

[2]The single edge restriction is for notational conveniences to follow; if a given neural model does have more than one edge between adjacent nodes, we can replace duplicate edges with 2-length paths through dummy identity nodes to satisfy this requirement without affecting its semantics.

Our approach is three-fold: first we employ abstractions of sets of paths as the localization and influence quantification instrument; second, we discover influential patterns with a greedy search procedure that refines abstract patterns into more concrete ones, keeping the influence high; and third, we consider the collection of influence patterns from every word in a sentence to the quantity of interest. We begin with the pattern abstraction:

**Definition 3 (Multi-partite pattern)** *A multi-partite pattern $\pi$ is a sequence of nodes $[\pi_1, \pi_2, \cdots, \pi_{-1}]$ such that for any pair of nodes $\pi_i, \pi_{i+1}$ adjacent in the sequence (not necessarily adjacent in the graph), there exists a path from $\pi_i$ and $\pi_{i+1}$.*

A pattern $\pi$ abstracts a set of paths, written $\gamma(\pi)$ that follow the given sequence of nodes but are free to traverse the graph between those nodes in any way. Interpreting paths and patterns as sets, we define $\gamma(\pi) \stackrel{\text{def}}{=} \{p \subseteq \mathcal{P} : \pi \subseteq p\}$ where $\mathcal{P}$ is the set of all paths from $\pi_1$ to $\pi_{-1}$. If every sequence-adjacent pair of nodes is directly connected then the pattern abstracts a single path.

**Definition 4 (Pattern influence)** *Given a computation graph and a DoI $\mathcal{D}$, the influence of a multi-partite influence pattern $\pi$, written $\mathcal{I}(\mathbf{x}, \pi)$ is the total influence of all the paths abstracted by the pattern:*

$$\mathcal{I}(\mathbf{x}, \pi) \stackrel{\text{def}}{=} \sum_{p \in \gamma(\pi)} \chi(\mathbf{x}, p) = \mathbb{E}_{\mathbf{z} \sim \mathcal{D}(\mathbf{x})} \prod_{i=1}^{-1} \frac{\partial \pi_i(\mathbf{x})}{\partial \pi_{i-1}(\mathbf{x})}$$

Also note that influence of individual paths may be positive or negative so cancellation in the influence of a pattern which aggregates paths is possible.

**Computation Graphs for BERT** A given DNN can be expressed by many computational graphs. For computational and interpretability reasons, an ideal graph would contain as few nodes and edges as possible while exposing structures of interest. For BERT in particular, we propose *embedding-level graph* $\mathcal{G}_e$ corresponding to the nodes and edges shown in Fig. 1 (left) to explain how the influence of input embeddings flow from one Transformer layer to another and to the eventual prediction of [MASK]; and *attention-level graph* $\mathcal{G}_a \supset \mathcal{G}_e$ that additionally includes head nodes as in Fig. 1 (right), a finer decomposition to demonstrate how influence from the input embedding flows through the attention block within each layer. BERT's semantics are modeled using a computational graph's activation functions which we omit here.

As the attention level graph contains a superset of the nodes of the embedding level graph, we can interpret embedding level patterns as abstracting paths in both the embedding level graph and the attention level graph. Furthermore, a concrete path in $\mathcal{G}_e$ is a pattern in $\mathcal{G}_a$ as it contains $\mathcal{G}_a$-non-adjacent nodes and thus abstracting multiple paths in $\mathcal{G}_a$. For a given pattern $\pi$ of $\mathcal{G}_e$ we can thus write $\gamma_a(\pi)$ as the set of paths it abstracts in $\mathcal{G}_a$ with:

$$\gamma_a(\pi) \stackrel{\text{def}}{=} \bigcup_{p \in \gamma_e(\pi)} \gamma_a(p)$$

**Guided Pattern Refinement(GPR)** Instead of enumerating the path space of $\mathcal{P}$ for discovering influential paths, we approximate a search by greedily refining patterns while maximizing their influence. Starting with sources and target nodes $s$ and $t$ along with a pattern $\pi^0 = \{s, t\}$ representing all paths between $s$ and $t$, we construct $\pi^1$ by adding a node from a *guiding set $E^0$* that maximizes the influence of the resulting pattern. At the first iteration and subsequently, the guiding set defines a cut of the (multi-partite) graph between two sequence adjacent nodes (initially just $s$ and $t$). The procedure is repeated with additional refinement. At iteration $i + 1$, a guiding set $E^i$ defines a cut between nodes $s^i$ and $t^i$ while the cut node that refines the pattern to maximal influence is selected:

$$\pi^{i+1} \stackrel{\text{def}}{=} \pi^i[s^i, t^i \setminus s^i, e^i, t^i]$$
$$e^i \stackrel{\text{def}}{=} \arg \max_{e^i \in E^i} \mathcal{I}(\mathbf{x}, \pi^i [s^i, t^i \setminus s^i, e^i, t^i])$$

Above, $\pi[a, c \setminus a, b, c]$ denotes the pattern $\pi$ in which sequence adjacent nodes $a, c$ are replaced with $a, b, c$, in their position in the sequence.

Repeating the procedure for some number of steps or until some stopping criterion is reached produces a sequence of patterns with decreasing abstraction: $\gamma(\pi^{i+1}) \subseteq \gamma(\pi^i)$. Once a pattern is produced that abstracts a single path, no more refinement can be done though it might not be desirable to continue refinement until that point for interpretability reasons. Also, the choice of guiding sets $E^i$ at each iteration can have an impact on the resulting patterns both in terms of their influence significance and computational requirements of iteration. Smaller sets require fewer options to enumerate but are likely to lead to less influential patterns.

In our experiments we employ a layer-ordered strategy for the embedding-level pattern refinement and then refine the resulting pattern in the attention-level graph. In the embedding-level analysis, at iteration $i$, we focus on layer $i$. The guiding set $E^i$ is the cut:

$$(\text{embedding-level guiding set}) \quad E^i \stackrel{\text{def}}{=} \left\{ h_j^l \right\}_j$$

The refinement thus proceeds for $L$ iterations (the input layer can be skipped). If the input node is denoted as $x$ and the quantity of interest is denoted as $q$, the refinement process results in a pattern $\pi^e \stackrel{\text{def}}{=} \left\{ x, h_{j_1}^1, h_{j_2}^2, \cdots, h_{j_L}^L, q \right\}$ where $j_i$ are indices designating which embeddings at each level $i$ the abstracted paths traverse.

The attention-level refinement starts with the embedding-level pattern $\pi^e$ and exposes the attention heads to cut the flow of influence in that starting pattern, also in order of the layers. At iteration $i$, the cut $E^i$ is:

$$(\text{attention-level guiding set}) \quad E^i \stackrel{\text{def}}{=} \left\{ a_{j_i}^{i,k} \right\}_k \cup \left\{ s_{j_i}^i \right\}$$

That is, the cut separates embedding nodes $h_{j_i}^i$ and $h_{j_{i+1}}^{i+1}$ with the attention heads $\left\{ a_{j_i}^{i,k} \right\}_k$ and a skip edge modeled as a node $s_{j_i}^i$. As the attention-level analysis refines the embedding-level analysis, the produced attention-level pattern $\pi^a$ abstracts a strict subset of the paths of the attention-level graph that the embedding-level pattern $\pi^e$ abstracts. That is, $\pi^e \subset \pi^a$ and therefore $\gamma_a(\pi^e) \supset \gamma_a(\pi^a)$.

In our experiments, we perform GPR independently for each input word, and refine with most positively influential cut nodes for positively influential words ($g_q(\mathbf{x}_i) \geq 0$)) but refine with the most negatively influential cut nodes for negative ($g_q(\mathbf{x}_i) < 0$) words. In the following section, we use $\pi_i$ as the extracted patterns for individual input word $i$, $\Pi$ as the set of patterns for all words, and $\Pi_+$ as the set of patterns for all positively influential words. Both terms may be further decorated by $a$ or $e$ to denote attention-level or embedding-level results.

## 4 EVALUATION

We apply GPR to discover BERT patterns on the level of embedding and attention. We begin with a summary of the linguistic tasks, datasets, models, and hyper-parameters. We evaluate patterns for their sparsity and sufficiency in Sec. 4.1 as measured by the metrics of *concentration* and *compression accuracy* (Lu et al., 2020), respectively. Finally, in Sec. 4.2 we visualize example patterns and discuss how they help explain contextualization of SVA in BERT.

**Tasks.** We consider two linguistic tasks: subject-word agreement (SVA) and reflexive anaphora (RA). We explore different forms of sentence stimuli in each task: object relative clause (Obj.), subject relative clause (Subj.), within sentence complement (WSC), and across prepositional phrase (APP) in SVA; number agreement (NA) and gender agreement (GA) in RA. SVA and RA datasets (Marvin & Linzen, 2018; Lin et al., 2019) are evaluated with MLM in a same way as prior work (Goldberg, 2019). We sample 200 sentences evenly distributed across different sentence types (e.g. singular/plural subject & singular/plural intervening noun) with a fixed sentence structure from each task; sentence length and the word types in each position are consistent across samples. Examples of each task are found in Appendix A.

**QoI and Distributional Influence.** We use the same QoI from Lu et al. (2020) where $q(\mathbf{y}_m) \stackrel{\text{def}}{=} y_{m,correct} - y_{m,wrong}$, e.g. $y_{m,\text{IS}} - y_{m,\text{ARE}}$ for `she [MASK] happy`. We select $\mathcal{D}$ as an uniform distribution over a linear path from $\mathbf{x}_b$ to $\mathbf{x}$ in the input space for each word with the baseline $\mathbf{x}_b$ defined as the the input embedding of `[MASK]`; we view it as a neutral word with no information.

**Model.** We evaluate our methods with a BERT model with $L = 6, A = 8$, referred hereby as BERT$_{\text{SMALL}}$, of Turc et al. (2019) instead larger models such as BERT$_{\text{BASE}}$ used in the original BERT

| Task | $lg|\mathcal{P}^*|$ | | $C^{\Pi^*}_+$ | | $C^{\Pi^*}_-$ | | $\frac{|\gamma_*(\pi^*_+)|}{|\mathcal{P}^*|}$ | | acc.$(\pi^*_+)$ | | acc.(rand. $*$) | | acc.(ori.) |
|---|---|---|---|---|---|---|---|---|---|---|---|---|---|
| | $e$ | $a$ | $e$ | $a$ | $e$ | $a$ | $e$ | $a$ | $e$ | $a$ | $e$ | $a$ | |
| **SVA** | | | | | | | | | | | | | |
| Obj. | 14.4 | 27.6 | 0.34 | 0.22 | 0.29 | 0.16 | 0.06 | 2.0e-7 | 0.99 | 0.74 | 0.50 | 0.51 | 0.96 |
| Subj. | 14.4 | 27.6 | 0.30 | 0.18 | 0.31 | 0.17 | 0.09 | 2.0e-7 | 0.74 | 0.56 | 0.50 | 0.49 | 1.00 |
| WSC | 13.8 | 27.0 | 0.27 | 0.16 | 0.38 | 0.24 | 0.03 | 1.8e-7 | 1.00 | 0.69 | 0.52 | 0.49 | 1.00 |
| APP | 14.4 | 27.6 | 0.33 | 0.18 | 0.31 | 0.15 | 0.06 | 1.6e-7 | 0.92 | 0.62 | 0.55 | 0.60 | 1.00 |
| **RA** | | | | | | | | | | | | | |
| NA | 14.4 | 27.6 | 0.31 | 0.18 | 0.32 | 0.18 | 0.04 | 1.4e-7 | 0.68 | 0.49 | 0.54 | 0.52 | 0.83 |
| GA | 15.4 | 28.6 | 0.22 | 0.14 | 0.23 | 0.15 | 0.02 | 1.4e-7 | 0.76 | 0.72 | 0.65 | 0.55 | 0.73 |

Table 1: Pattern sufficiency, sparsity, and related metrics on various linguistic tasks. Metrics are shown in the 1st row while the 2nd row indicate graph levels: $*$ denotes $e$ or $a$, corresponding to abstracted embedding-level patterns $\pi^e$ or attention-level patterns $\pi^a$, respectively. $lg|\mathcal{P}^*|$: natural log of the number of possible paths; $C^{\Pi^*}_+$, $C^{\Pi^*}_-$: positive/negative concentrations; $\frac{|\gamma_*(\pi^*_+)|}{|\mathcal{P}^*|}$: percentage of paths in the abstracted patterns over the total number of paths; acc.$(\pi^*_+)$: the compressed accuracy of abstracted patterns; acc.(rand. $*$): the compressed accuracy of randomly compressed models; acc.(ori.): the accuracy of the original model BERT$_{\text{SMALL}}$.

paper (Devlin et al., 2019) because 1) we find BERT$_{\text{BASE}}$ is not significantly better than BERT$_{\text{SMALL}}$ in the tasks of interests as shown in Appendix A, 2) when approximating the expectation in Def. 2 with finite points, we find more than 2000 samples are required in BERT$_{\text{BASE}}$ for an acceptable margin, while 50 samples suffices for BERT$_{\text{SMALL}}$ (see Appendix. B.1), and 3) visualizations of abstracted patterns from BERT$_{\text{SMALL}}$ are easier on human interpreters.

### 4.1 QUANTITATIVE ANALYSIS

Recall that $\pi^e$ and $\pi^a$ denote the abstracted pattern returned by GPR with an embedding-level graph and an attention-level graph, respectively. The quantitative evaluation in this section aims to verify that influential patterns in BERT are 1) sparse: inputs influence the QoI largely through $\pi^e$ or $\pi^a$ and 2) sufficient: BERT retains high task accuracy if only $\pi^e$ or $\pi^a$ are evaluated at inference time. Firstly, we introduce *concentration* to evaluate sparsity:

**Definition 5 (Concentration)** *Given an input* $\mathbf{x}$ *with $N$ words/embeddings, their distributional influences $g_q(\mathbf{x})$ and pattern influences $\mathcal{I}(\mathbf{x}, \pi_i)$ for a set of patterns $\Pi = \{\pi_i\}_i$, the concentration of the positive (and negative) pattern influence $C^{\Pi+}$ (and $C^{\Pi-}$) are the patterns' share of positive (or negative) influences as compared to the total positive (or negative) distributional influence:*

$$C^{\Pi+} \overset{\text{def}}{=\joinrel=} \frac{\sum_i^N \{\mathcal{I}(\mathbf{x}, \pi_i)_i * \mathbb{I}[\mathcal{I}(\mathbf{x}, \pi_i)_i \geq 0]\}}{\sum_i^N \{g_q(\mathbf{x})_i * \mathbb{I}[g_q(\mathbf{x})_i \geq 0]\}}, C^{\Pi-} \overset{\text{def}}{=\joinrel=} \frac{\sum_i^N \{\mathcal{I}(\mathbf{x}, \pi_i)_i * \mathbb{I}[\mathcal{I}(\mathbf{x}, \pi_i)_i < 0]\}}{\sum_i^N \{g_q(\mathbf{x})_i * \mathbb{I}[g_q(\mathbf{x})_i < 0]\}}$$

To evaluate sufficiency, we employ a compression study previously used to verify other explanation devices (Dabkowski & Gal, 2017; Ancona et al., 2018; Leino et al., 2018; Lu et al., 2020). For each example, we *compress* BERT down to a specific pattern: we only retain the nodes from $\Pi^e_+$ (or $\Pi^a_+$) while replacing all other nodes, layer by layer. Starting from the first layer, the embedding nodes not in $\Pi^e_+$ (or the attention/skip connection nodes not in $\Pi^a_+$) are replaced by the embedding of [MASK] (or zero vectors for attention/skip nodes), while the nodes in $\Pi^e_+$ (or $\Pi^a_+$) remain untouched. The retained and replaced node together are forward passed to the next layer using the original model parameters until a new set of nodes needs to be retained or replaced. We then compare the accuracies of predicting labels of [MASK] between the original model and the compressed model, the former of which we refer to as *compressed accuracy*.

**Explanations of Results.** Concentration and compressed accuracy of $\pi^e$ and $\pi^a$ are shown in Table 1 using the setup in Sec. 4. Replacing $*$ in the second row with $e$ or $a$ corresponds to the results for the abstracted embedding-level patterns $\pi^e$ and the attention-level patterns $\pi^a$, respectively.

**Sparsity with concentration.** Per the gradient chain rule, the total influence of all individual paths from the input to QoI equals to the distributional influence, therefore, $0 < C^{\Pi+}, C^{\Pi-} < 1$. As

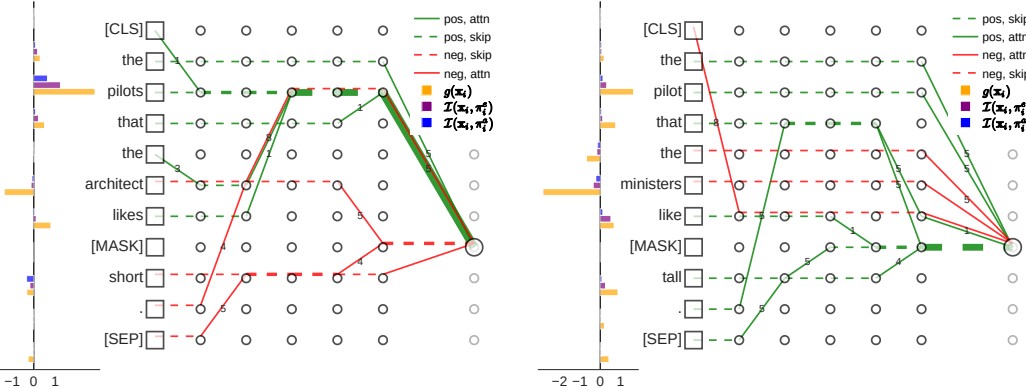

(a) plural subject + singular intervening noun(PS)   (b) singular subject + plural intervening noun(SP)

Figure 2: Significant patterns $\pi^a$ extracted by GPR from the attention-level graph for task *SVA across Object Relative Clauses* (Goldberg, 2019; Marvin & Linzen, 2018), in two exmaples with attractors. Left: bar plots of the distributional influence $g(\mathbf{x}_i)$(yellow), $\mathcal{I}(\mathbf{x}_i, \pi_i^e)$ (purple) and $\mathcal{I}(\mathbf{x}_i, \pi_i^a)$ (blue) for each word at position $i$. Right: significant patterns $\pi_i^a$ from each input word at position $i$ to quantity of interest (verb number correctness). The square nodes denote the input embeddings and circles denote internal contexualized embeddings. Dashed lines correspond to skip connection in the attention block while solid lines correspond to connection through (any) attention heads. Attention connections with high influence flow are marked with the corresponding attention head number (ranging from 1 to 8). Line colors represent the sign of influence (red as negative and green as positive).

shown in the first two columns of Table 1, there are about $\exp 14$ and $\exp 27$ individual patterns in the embedding and attention-level graphs, respectively. However, the abstracted patterns (shown by the 3$^{\text{rd}}$ to 6$^{\text{th}}$ columns) account for a large portion of both positive and negative influence across all tasks. The embedding level ($* = e$) abstracted pattern contributes around 30% of the total influence indicating that the concept is concentrated to individual contextualized embeddings in each layer, instead of dispersed to many words. Zooming in on the attention-level ($* = a$), concentration suggests that between the contextualized embeddings of adjacent layers, influence is also more concentrated to either one attention head or the skip connection.

**Sufficiency with model compressed accuracy.** We show the original accuracies of the model on different tasks in the last column. The compressed model retains a high accuracy (as shown in 9$^{\text{th}}$ and 10$^{\text{th}}$ columns) with a tiny portion of the models (shown in 7$^{\text{th}}$ and 8$^{\text{th}}$ columns) retained. As a comparison, we denote the compressed accuracy with random patterns in the 11$^{\text{th}}$ and 12$^{\text{th}}$ columns, which compresses the model by retaining the same number of nodes as $\pi_+^e$ but has a performance close to 50%, effectively a random guess; randomly chosen patterns of the same size do not abstract the concept at all.

### 4.2 EXPLAINING CONTEXTUALIZATION OF SVA ACROSS OBJECT RELATIVE CLAUSE

In this section, we explain contextualization between internal representations of BERT by visualizing the significant patterns $\pi^e$ and $\pi^a$ found by GPR for two examples of SVA across object relative clauses as seen in Figure 2. Results on other tasks are included in the Appendix. B.2.

First we observe that in both sentences of Figure 2, both words in the subject phrase ("the" and the nouns) exert a positive input influence on the correct prediction of the verb, and the intervening noun(attractor) exerts negative influence, which is also true for both $\mathcal{I}(\mathbf{x}, \pi_i^e)_i$ and $\mathcal{I}(\mathbf{x}, \pi_i^a)_i$.

**"Copy and Transfer"** We observe many horizontal dashed lines in Figure 2, indicating significant influence flows through layers at the same word position using skip connections. Zooming in on $\pi_i^e$ and $\pi_i^a$, we observe that the subject phrase travels through skip connections across the lower layers, and only through attention head 5 in the last layer. This "copy and transfer" procedure indicates that BERT mostly picks up the signal from the subject input embedding without much contextualization,

however, exactly how it overcomes(or not) the comparable signals from the attractors is explained the next section. In addition, we speculate that the reason that attentions can be effectively pruned without compromising the performance in prior works (Michel et al., 2019), is that some concepts does not travel through attention block at all: they are simply "copied" to the next layer through the skip connections. In Appendix B.2 we observe that all the above conclusions are also prevalent in other tasks (such as the contextualization of propositions in WPP task), though different heads might be used for the "transfer" operations in different tasks.

**The Role of `that`** Comparing two sentences of Figure 2 , we do observe that the influence from the singular subject is weaker than that of the plural subject, especially compared to the negative influence from attractors. The key difference is that `that` behaves differently for singular and plural subject. `that` in Figure 4a behaves as a singular noun (since `that` also means a singular pronoun in English), flowing through the same straightforward pattern as the subject (skip connections + attention head 5); `that` in Figure 2a, however, behaves more like a grammatical marker(relativizer): the pattern from `that` travel from itself to the subject in the second to last layer through a different attention head. We speculate that `that` in plural subject sentences encodes the syntactic boundary of the clauses and help identify the main subject and ignore the intervening noun. As a result, attractors in PS have smaller negative influence, compared to the high negative influence from attractors in SP (a similar comparison is also observed in PP and SS cases). This discrepancy in the behavior of `that` also corroborates lower SVA accuracy in SP case than in PS case (See Appendix A). We observe this difference consistently across all instances as shown in an aggregated visualization in Appendix B.2, in other tasks as well.

## 5  RELATED WORK

Previous work has shown the encoding of syntactic dependencies such as subject verb agreements(SVA) in RNN Language models (Linzen et al., 2016), as well as the explanations for such encoding(Hupkes et al., 2018; Lakretz et al., 2019; Jumelet et al., 2019). More extensive work has since been done on transformer-based architectures such as BERT(Devlin et al., 2019). Diagnostic classifiers trained on output and internal representations discover that BERT encodes many types of linguistic knowledge(Elazar et al., 2020; Hewitt & Liang, 2019; Tenney et al., 2019a;b; Jawahar et al., 2019; Klafka & Ettinger, 2020; Liu et al., 2019; Lin et al., 2019), ranging from syntactic concepts to more complicated semantic ones. Goldberg (2019) discovers that SVA and RA in complex clausal structures is better represented in BERT compared to an RNN model. This is partially explained by (Reif et al., 2019; Hewitt & Manning, 2019) which show that contextual embeddings in BERT can encode syntactic structures hierarchically comparable to those represented in a dependency tree. However all these analyses are done on frozen contextual embedding layers; the exact causal mechanism a concept is encoded from input to output is not explored.

Another line of work in interpreting BERT concerns analyzing the self-attention weights of BERT(Clark et al., 2019; Vig & Belinkov, 2019; Lin et al., 2019), where attention heads are found to have direct correspondences with specific dependency relations. However, attention weights as interpretation devices has been controversial(Serrano & Smith, 2019), and empirical analysis has shown that attention can be perturbed or pruned while having the same or even better performance(Kovaleva et al., 2019; Michel et al., 2019; Voita et al., 2019). More importantly, our work demonstrate that attention mechanisms are only part of BERT computation graph, with each attention block complemented by additional architecture such as dense layer and skip connections. The strong influence passing through skip connections also corroborates the findings of Brunner et al. (2020) which find input tokens mostly retain their identity. Besides pruning attentions, other works(Prasanna et al., 2020; Sanh et al., 2019; Jiao et al., 2019) also show that BERT is overparametrized and can be greatly compressed. Our work to some extent corroborates that point by pointing to the sparse gradient flow, while employing model compression only to verify the sufficiency of the extracted patterns.

Recent work introducing influence paths (Lu et al., 2020) offers another form of explanation. Lu et al. (2020) decomposed the attribution to path-specific quantities localizing the implementation of the given concept to paths through a model. The authors demonstrated that for LSTM models, a single path is responsible for most of the input-output effect defining SVA, and explored the effects

of unhelpful nouns which showed negative influence on SVA. We describe the limitations of this methodology when applied to BERT in Sec. 3.

## 6 CONCLUSION

We have demonstrated how to use multi-partite influence patterns to localize a DNN model's handling of a concept of interest and along with a pattern refinement method we how BERT handles subject-verb number agreement and reflexive anaphora. We quantitatively validated the sufficieny and sparsity of influence patterns in BERT by way of compression experiments and the influence concentration of discovered patterns. We qualitatively and visually demonstrated BERT's contextualization in the two tasks using our methodology. Our formalism and methods are general enough to apply to the analysis of other aspects of BERT and other models.

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

## A   APPENDIX: INTRODUCTION TO LINGUISTIC TASKS

| Task | Type | Example | BERT Small | BERT Base |
|---|---|---|---|---|
| SVA | | | | |
| Object Relative Clause | SS SP PS PP | the author that the guard likes [MASK(is/are)] young | 1 0.92 0.9 1 | 1 0.96 0.98 1 |
| Subject Relative Clause | SS SP PS PP | the author that likes the guard [MASK(is/are)] young | 1 1 1 1 | 1 0.96 0.98 1 |
| Within Sentence Complement | SS SP PS PP | the mechanic said the author [MASK(is/are)] young | 1 1 1 1 | 1 1 1 1 |
| Across Prepositional Phrase | SS SP PS PP | the author next to the guard [MASK(is/are)] young | 1 1 0.98 1 | 0.99 0.98 0.98 1 |
| Reflexive Anaphora | | | | |
| Number Agreement | SS SP PS PP | the author that the guard likes hurt [MASK(himself/themselves)] | 0.66 0.66 0.83 1 | 0.6 0.74 0.83 0.96 |
| Gender Agreement | MM MF FF FM | some wizard who can dress our man can clean [MASK(himself/herself)] | 0.78 0.32 1 0.8 | 1 0.96 0.9 0.66 |

Table 2: Example of each agreement task and their performance on two BERT models, first 5 tasks are sampled from Marvin & Linzen (2018), the last task is sampled from dataset in Lin et al. (2019), all datasets are constructed as an MLM task according to Goldberg (2019).

# B    APPENDIX: EXPERIMENT DETAILS

## B.1    CONVERGENCE CHECK

When the DoI of *distributional influence* $g_q(\mathbf{x})$ is a uniform distribution on a linear path from a baseline input $\mathbf{x}_b$ to the target input $\mathbf{x}$, the *completeness* (Sundararajan et al., 2017) axiom shows $q(f(\mathbf{x})) - q(f(\mathbf{x}_b)) = \sum_i x_i g_q(\mathbf{x})_i$, where $q$ is the selected Quantity of Interest. However, when summation is used to approximate the expectation in practice, the RHS of the *completeness* axiom does not always converges to the LHS easily. In Fig. 3, we plot the percentage of difference $[q(f(\mathbf{x})) - q(f(\mathbf{x}_b)) - \sum_i x_i g_q(\mathbf{x})_i]/(q(f(\mathbf{x})) - q(f(\mathbf{x}_b)))$ against the resolution, the number of samples drawn from the distribution in the summation. Due to the limit of our GPU memory ( 12GB) and the computational cost, we find the maximum number of batched samples to be 50. Therefore, BERT$_{\text{SMALL}}$ has lower approximation error compared to BERT$_{\text{BASE}}$. The much harder approximation of the larger BERT model is likely due to the complicated decision boundaries of larger BERT, masking the output sensitive to small perturbations.

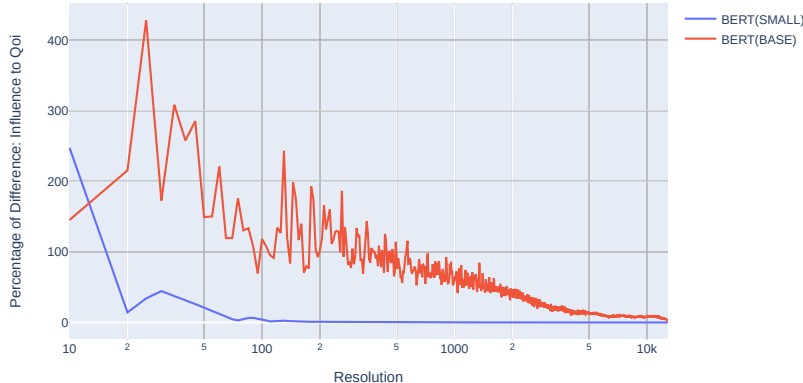

Figure 3: Convergence Analysis for Calculating the Distributional influence(IG) for SVA Across Object Relative Clauses for BERT$_{\text{SMALL}}$(used in this paper) BERT$_{\text{BASE}}$ from Devlin et al. (2019) perturbations. X-axis is the number of samples used to approximate the influence, Y-axis is the percentage of deviation from approximation to true influence value.

## B.2    AGGREGATED INFLUENCE GRAPHS FOR ALL TASKS

In this section, we show the an aggregated visualization across all examples by superimposing the visualization of individual instances as the ones in Figure 2, while adjusting the line width to be proportional to the frequency of flow across all examples. The words within parenthesis represent one instance of the word in that position.

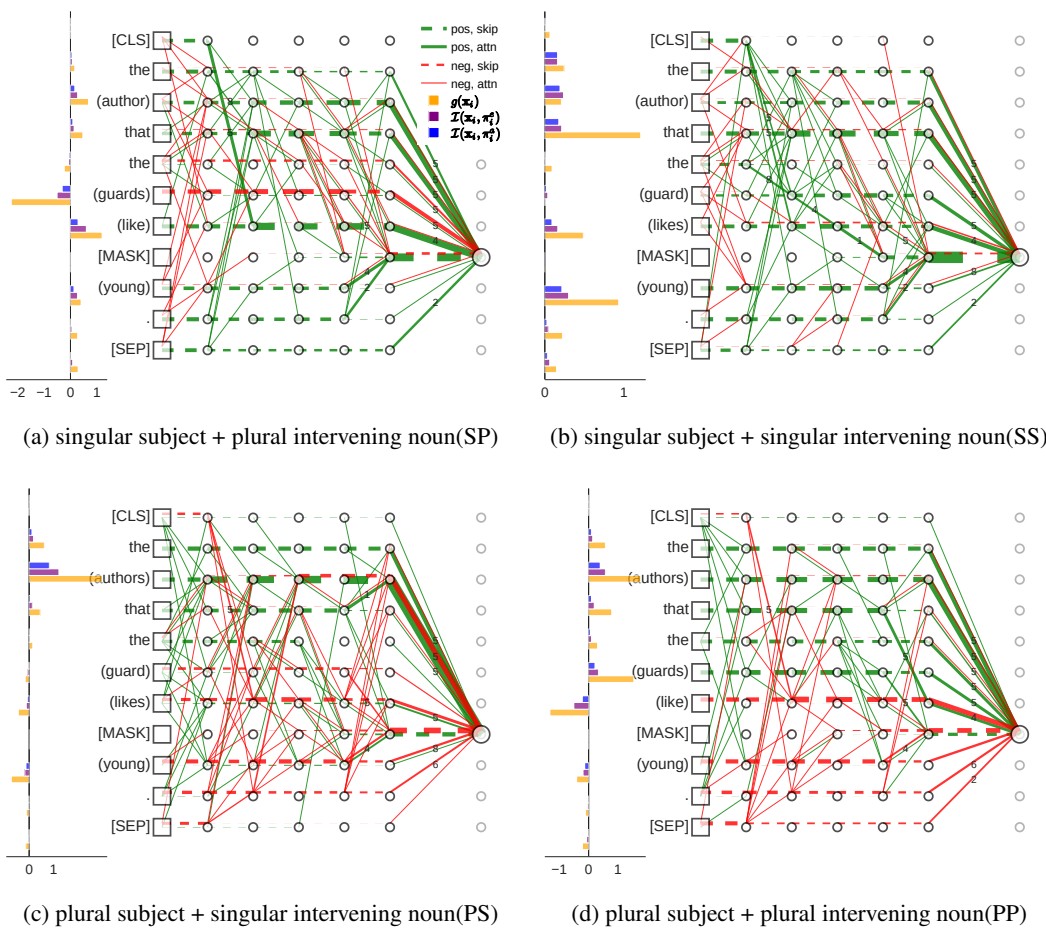

(a) singular subject + plural intervening noun(SP)

(b) singular subject + singular intervening noun(SS)

(c) plural subject + singular intervening noun(PS)

(d) plural subject + plural intervening noun(PP)

Figure 4: SVA Across Object Relative Clause.

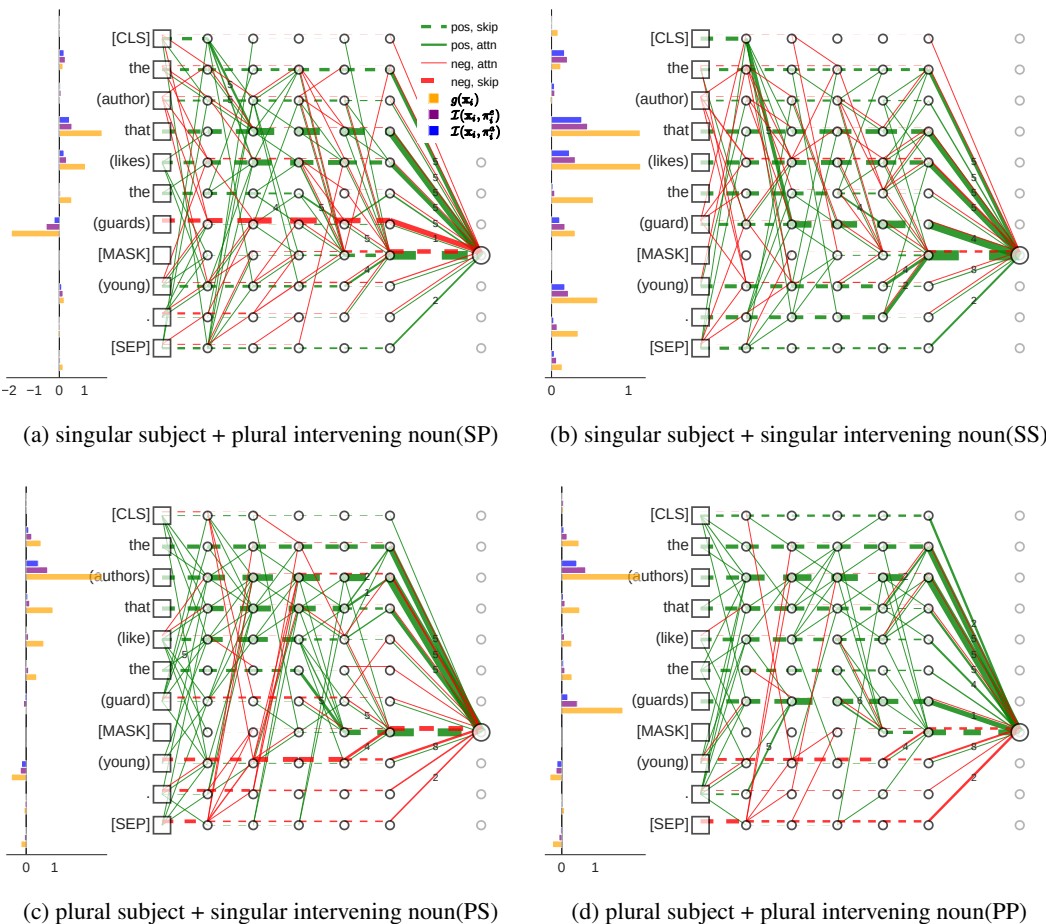

(a) singular subject + plural intervening noun(SP)

(b) singular subject + singular intervening noun(SS)

(c) plural subject + singular intervening noun(PS)

(d) plural subject + plural intervening noun(PP)

Figure 5: SVA Across Subject Relative Clause.

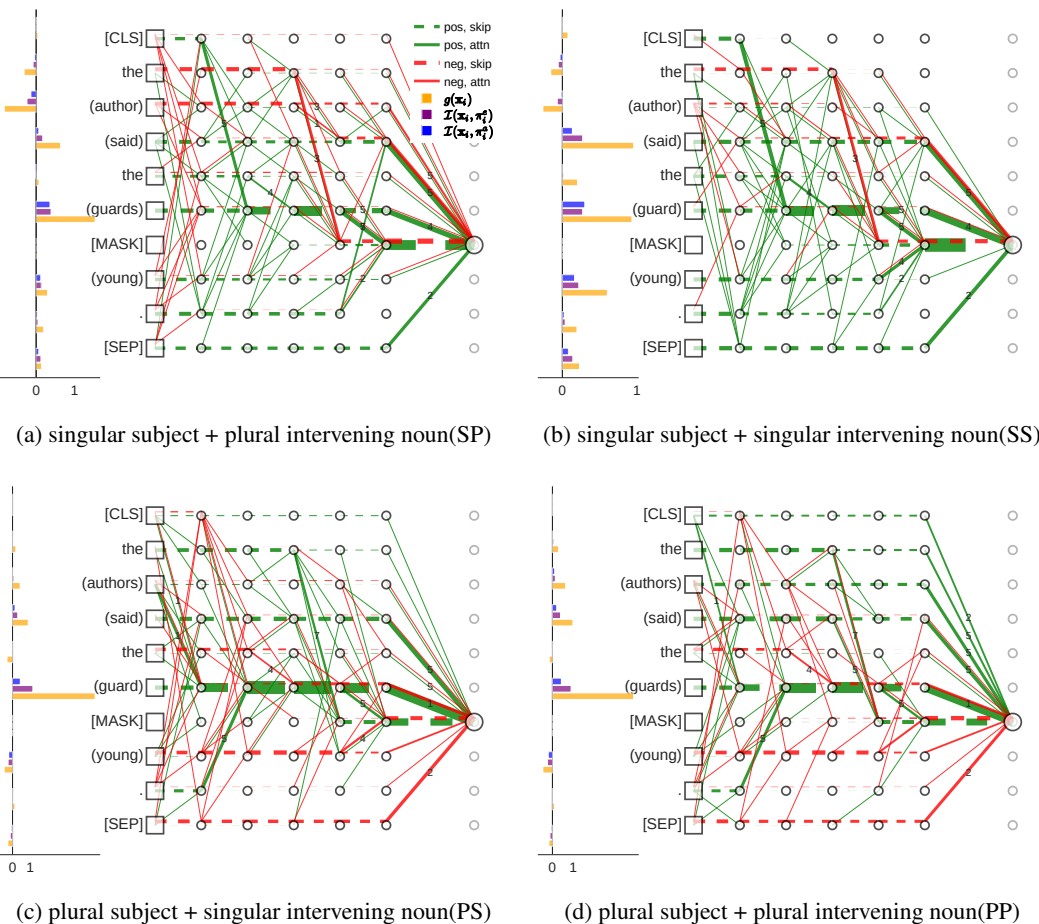

(a) singular subject + plural intervening noun(SP)

(b) singular subject + singular intervening noun(SS)

(c) plural subject + singular intervening noun(PS)

(d) plural subject + plural intervening noun(PP)

Figure 6: SVA Within Sentence Complements.

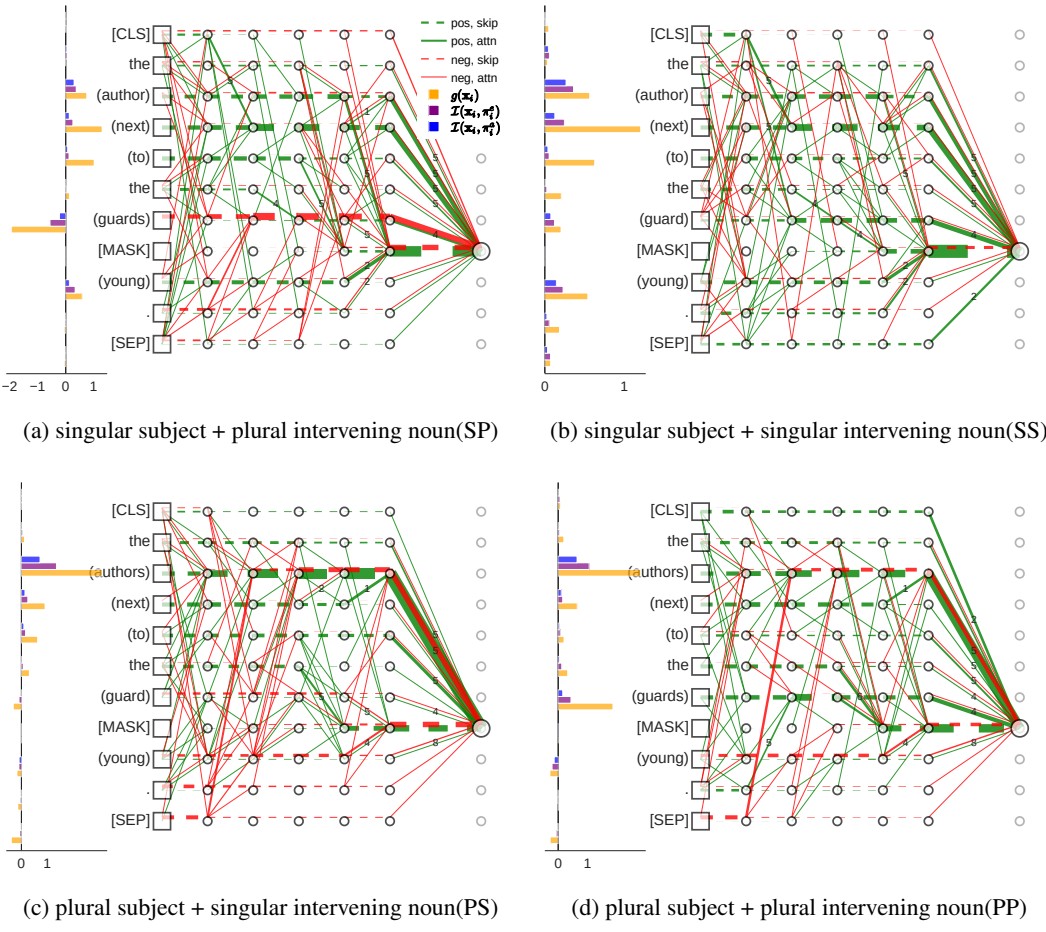

(a) singular subject + plural intervening noun(SP)

(b) singular subject + singular intervening noun(SS)

(c) plural subject + singular intervening noun(PS)

(d) plural subject + plural intervening noun(PP)

Figure 7: SVA Across Prepositional Phrase.

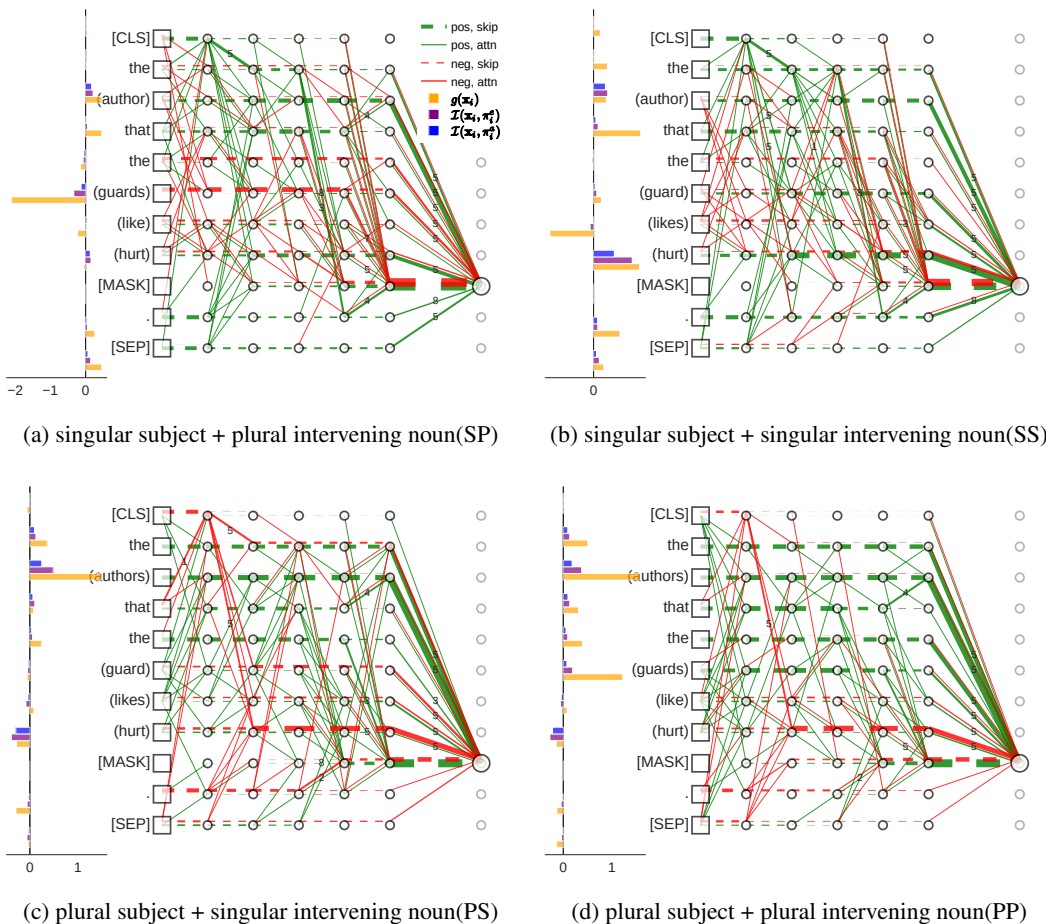

(a) singular subject + plural intervening noun(SP)

(b) singular subject + singular intervening noun(SS)

(c) plural subject + singular intervening noun(PS)

(d) plural subject + plural intervening noun(PP)

Figure 8: RA: Number Agreement

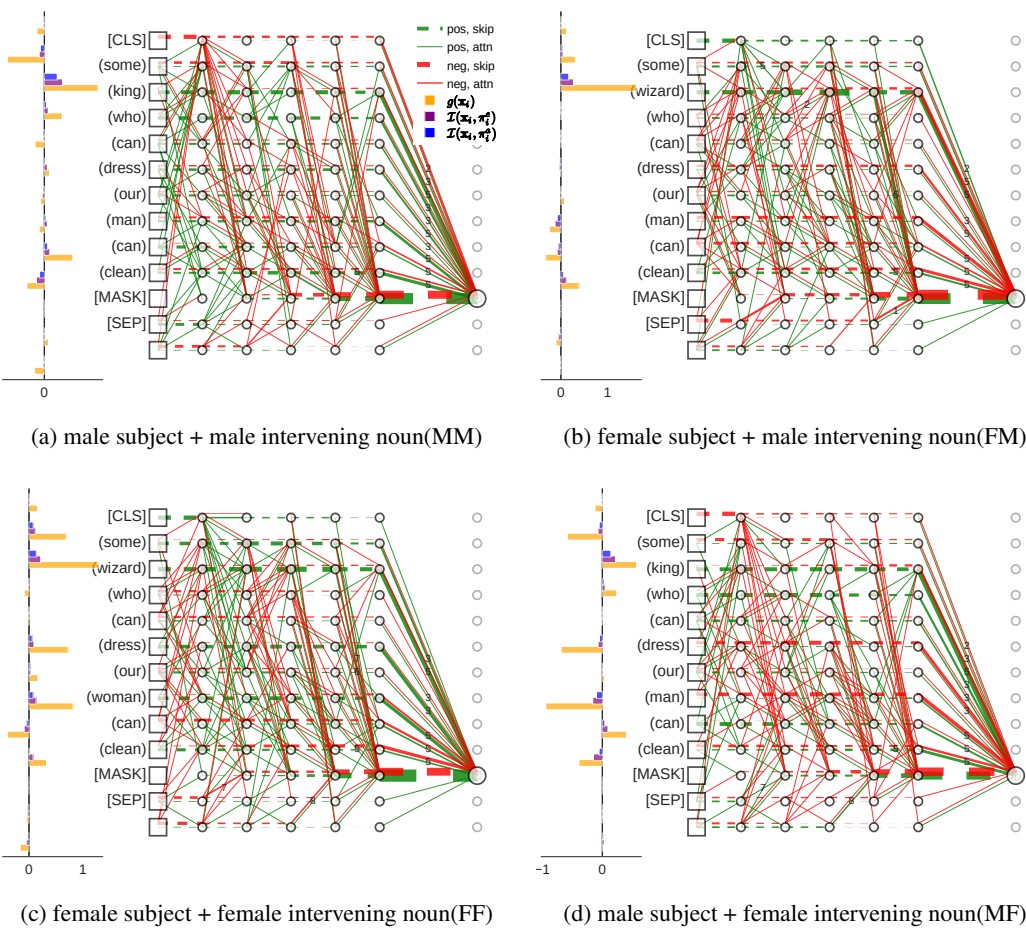

(a) male subject + male intervening noun(MM)

(b) female subject + male intervening noun(FM)

(c) female subject + female intervening noun(FF)

(d) male subject + female intervening noun(MF)

Figure 9: RA: Gender Agreement

