# OpenReview forum: "ABSTRACTING INFLUENCE PATHS  FOR EXPLAINING (CONTEXTUALIZATION OF) BERT MODELS"
_ICLR.cc/2021/Conference — Reject_

### Official Review · AnonReviewer3 · 2020-10-24
**Promising explanation method; novelty is a bit limited and some parts are a bit confusing**

**Rating:** 6
**Confidence:** 4

**Review:**

#### Summary

This paper proposes a new explanation method of DNNs by applying Distributional Influence analysis on the proposed Multi-partite patterns. A Multi-partite pattern $\pi$ is basically a partial path; it "abstracts" a set of paths $\gamma(\pi)$ which consists of the paths that include all edges in the pattern. The influence of $\pi$ is thus defined as the sum of path influences among $\gamma(\pi)$. The authors also propose Guided Pattern Refinement (GPR) algorithm to find the most influential patterns for a certain input word. It is a greedy algorithm that adds an edge per step to maximize the current pattern influence.

Although this method is in principle applicable to any DNNs, this paper is mostly focused on BERT. Experiments are done to evaluate and showcase the usefulness of the proposed method. The experiment results on Concentration (of influence) and Model Compression illustrated that the extracted patterns from BERT indeed capture our quantity of interest (QoI). The authors also utilize the extracted patterns to shed more light on BERT behaviors.


#### Strength

- This paper is targeted at BERT explanation, which is very important and valuable. Model explanation is gaining more and more attention, especially for BERT, given its dominant popularity and yet hardness in interpretations.
- The proposed explanation method is reasonable and it proved useful in the experiments. The Concentration and Compression results are promising (except that comparisons to other methods are not provided, which is a weakness). Also, the behaviors discovered and discussed are also interesting.


#### Weakness

- It looks like the proposed method is largely based on (Lu et al, 2020); the major difference is the introduction of Multi-partite patterns, which basically expands the objects of influence analysis from paths to patterns (partial paths). This doesn't look like a significant novelty.
- Several parts of the paper are very confusing to me, especially the background section (maybe because my primary research area is not model interpretation). I tried to read several references but still have many questions unsolved. I will put specific questions in the next section.


#### Questions & Suggestions
- Confusions
	- Section 2, Distribution Influence (paragraph): "Influence can quantify a concept ..." might be misleading. Influence is not quantifying a concept but quantifying the impact of an input concept (subject S/P) on an output concept (verb S/P), as stated in Definition 1.
	- Section 2, Definition 1: "... a DoI D(x) parameterized by an input, ..." Here "parameterized by an input" confused me a lot when I first read it. After checking references and the experiments, I guess you are referring to a distribution like the uniform distribution from original embedding to [MASK] embedding, where the original embedding is the "parameter". However, this sense is somewhat unusual... I always thought about "parameterize" like $\mu,\theta$ in Gaussian distributions and thus cannot understand. Also, I think the word "input" here is ambiguous: you have an "analysis input", which is $x$; you also have "model input", which are samples from $D(x)$. It might be better to explicitly distinguish them.
	- Section 2, Line 22-23: "... in our use cases DoI are distributed along a one dimensional path $c:[0,1]\to R^d$." This does not look valid mathematically; how can a distribution be over a function? I guess what you mean is that DoI are distributed over $c=\\{x_m+\alpha(x-x_m),\alpha\in[0,1]\\}$.
	- Section 3, Guided Pattern Refinement(GPR): "One could repeat the procedure until an individual path is produced though for reasons of interpretability, it might be preferable to reason about patterns instead of single paths." How are you actually extracting patterns? As I see in Figure 1 and 3, you are illustrating paths. Do you mean "repeat the procedure until an individual path, but (mentally) reason about patterns in the path"? It looks like so according to your experiments discussed in Table 1.

- Others
	- Section 4: (a minor one) It is kind of inconvenient for readers that you frequently refer to Figure 1 in your discussion, which is many pages away. Would it be better if you move or copy Figure 1 and put it together with Figure 3?


#### Typos
- Figure 1, Line 1: "GRP" -> "GPR"
- Page 3, Line 2: extra space
- Page 4, Def 4: in the middle term, $x$ -> $z$
- Page 4, Def 5, Line 4: $n_w,n_w$ -> $n_w,n_q$
- Page 7, Line 2: "retain a even", "a" -> "an"
- Page 7, Line -9: "In ," -> (In all?)

---

> ### Author Response · Authors · 2020-11-18
> **Response to Reviewer 3**
>
> We appreciate all the suggestions and questions provided by the reviewer.
>
>
> 1) Novelty concerns with [Lu et al 2020]
>
> Apart from the extension from path to pattern mentioned by the reviewer, we hereby highlight a few more differences: (1) since it requires exhaustive searching,  Lu et al. 2020 is not able to scale to BERT without the fine-grained algorithm GPR, not  even for $BERT_{small}$ (see Table 1 where more than e^20 paths for the attention-level graph)
> (2)  Lu et al 2020 analyzes LSTM where the basic nodes are gates. In this paper,  we introduce two different graph schemes, embedding level and attention level and demonstrate how to search for significant attention-level patterns from abstracted embedding level patterns.
>
> 2) Improving the readability
>
> We appreciate the reviewer for pointing out the readability issues. In the updated version, we have updated the writing and we summarize major changes of the introduction and the results section in our official comments to all reviewers. In addition, we have also updated the background section, where we add pointers to original BERT and self-attention papers. We also improve readability by 1) adding a better visualization (Fig 1 in the updated submission) of the BERT architecture to support our analysis, 2) explaining why we focus on MLM tasks,  and 3) explaining the DoI with more clear notations.
>
> The convergence analysis in the Appendix A shows how many points are required to approximate the expectation with summation in practice.
>
> 3) Response to other confusions:
>
> We thank the reviewer for the feedback on confusing terms and descriptions. We have updated the submission to resolve the confusion 1 - 3 in the background section. For confusion 4, we modify the description of GPR to show how to search for significant patterns. The difference between patterns and paths is that the former include all possible paths between nodes. So the edges in Figures do not  represent specific computation edges, but the specific attention-level nodes (dashed lines are skip connections, while solid lines marked with numbers are attention heads).
>
> 4) Position of Figure 1
>
> We have moved Fig 1(Fig 2 in the new submission) to the Evaluation section. We appreciate your pointing out the concern of extra burden for readers to have to go back to the beginning of the paper.

---

> > ### Comment · AnonReviewer3 · 2020-11-23
> > **Response to rebuttal**
> >
> > Thanks for the response and update on the paper draft! The confusion issues are mostly addressed in the updated version, which is great. I believe the readability is improved, but my overall ideas about novelties and contributions are not changed much, so I'll keep my score (or changing it from 5.7 to 6.3 if this is allowed :) )

---

### Official Review · AnonReviewer1 · 2020-10-24
**Potentially interesting, but not clear what the question is, why this method, or what answers we get**

**Rating:** 6
**Confidence:** 4

**Review:**

This paper analyzes functioning of BERT by identifying gradient-based influence paths to track flow of influence between model inputs and outputs. Such analysis using influence paths has been established in prior work, but the present paper expands on this work with definition of "multi-partite" patterns, and by introducing a method for identifying strongly influential paths in the model. The authors evaluate on existing datasets for studying subject-verb agreement and reflexive anaphora, and they report measures of concentration of the influence flows, and performance of the model after compression based on the identified influence paths.

Overall, I think there are certainly interesting analyses that could come out of this work, but the current paper does not provide a clear enough contribution to be ready for publication.

The reported results don't give us much to go on in terms of interpreting what we have learned from these analyses. The authors choose two measures: "concentration" of the influence paths, and drop in accuracy when compressing the model based on those paths. I'm not sure what we should be taking away from the concentration numbers, and the authors don't provide any substantive discussion of why this matters. There is also no meaningful baseline against which to compare these numbers. The second measure, accuracy of the compressed model, is a bit easier to interpret, in that it allows us to verify the extent to which critical information flow is indeed occurring within the identified paths, but while this verification serves to increase confidence in the method, it's not clear what it tells us about the functioning of the BERT model. (The authors also do not give a clear statement of what task these accuracies are in fact for, though I assume by default that they are referring to accuracy in assigning higher probability to the grammatical option over the ungrammatical option in the selected Marvin & Linzen datasets.) There is also no discussion of how we should interpret results from the attention-based versus the embedding-based influence paths.

Zooming out further, there isn't a clearly identifiable question being asked in this paper, and in particular there is no clear connection between the analysis method being used and the particular linguistic phenomena embodied by the chosen evaluation data. What question is being asked about the model's handling of these linguistic phenomena, and why is this method appropriate for asking it? What output of the analysis will be used as an answer to the key questions? These connections should be defined clearly and from the start. As it is, I'm unclear on what these results tell us about the chosen linguistic phenomena (or, conversely, why this set of phenomena was chosen to showcase utility of this method). The measures reported have some variation between sentence types -- but what does it mean that there is higher positive embedding influence concentration for sentences with subject relative clauses, or higher negative embedding influence concentration for within sentence complement sentences? What is the meaning of the accuracies dropping more for subject relative clauses and number agreement among the embedding influence paths, and all accuracies seemingly dropping for the attention influence paths?

In Section 4.2 there are a couple of observations about the influence paths that make some concrete connections to potential questions about syntactic dependencies in the chosen datasets. However, because no formal connection between these things has been established prior to this point (no clear question, no clear linking hypothesis between the analysis and the phenomena) this set of observations comes a bit out of the blue, and leaves us still without clear takeaways. In sum, I think that there is potential for interesting insights about these phenomena to come out of this analysis method, but the paper would benefit from much more clearly defined questions and linking assumptions.

---

> ### Author Response · Authors · 2020-11-18
> **Response to Reviewer 1**
>
> We appreciate all the suggestions and questions provided by the reviewer.
>
> We agree with the reviewer on the paper’s lack of key questions and connections between the results and those questions. We have made revisions in the updated submission to address those concerns, summarized in the following three topics:
>
> 1) What is the central question and how does our analysis answer elements of the question.
>
> We have edited the introduction by posing a group of central questions right in the beginning: “How is contextualization realized in BERT for grammatical concepts such as SVA and RA?, Specifically, can we identify sub-components of BERT that are a) sufficient for representing those concepts but also b) sparse enough to legibly show how BERT contextualizes the concept across layers and whether the contextualization follows correct grammatical rules?” We have also added more takeaways in the evaluation section, each subsection of which demonstrates results that answer part of the central question.
>
> Firstly, the compression results, clarified in the updated version as compressed accuracy, indeed refer to the task accuracy, e.g., the accuracy for assigning the correct verb over the wrong verb for SVA. High compressed accuracy indicates the abstracted patterns are sufficient in representing the concept in BERT, even though only a small portion of the model is retained.
>
> Secondly, the concentration results show the sparsity of the abstraction: since the input influence equal the sum of all path influences, the fact the abstracted pattern only accounts for a small number of total paths while account for a larger portion of the total influence, shows the efficacy of the GPR searching algorithm, as well as the sparsity and interpretability of the abstracted patterns.
>
> Finally, section 4.2, which shows a set of visualizations of abstracted patterns, gives a qualitative analysis of how influence patterns help us interpret how BERT contextualizes concepts like SVA.  We discover that a lot of contextualization happens via the skip connections, a previously overlooked element of the model, and that influence patterns can also tell us if the contextualization corresponds to the actual grammatical rules, exemplified by the contextualization of the relativizer “that”. However, results in this section are qualitative instead of a strict comparison with actual syntactic dependency parsing required for the tasks, which we leave for future work.
>
> Since our method is quite unique compared to prior approaches including probing classifiers and attention analysis, there isn’t a good baseline to compare with. However, we have done experiments with baseline patterns created by high attention weights in each layer and discovered the derived patterns are inconsistent and random across examples, also far worse compared to the influence-based patterns. We will include those results in the final version given more space.
>
> 2) Why this method is appropriate.
>
> In the updated version we have added more clarifications in the background to motivate the choice of our method. In summary, our method is appropriate for answering the central questions is that input influence as an explanation device is axiomatically justified and widely used in other areas such as images, but has not been applied to BERT, except in real open-source frameworks (https://captum.ai/tutorials/). In this paper, we base our analysis by searching for the subcomponents of BERT where significant influence flows.  We apply our methods to widely studied datasets and linguistic phenomena from Goldberg 2019  and Lin et al 2019, where the MLM setup enables easier definition of QoI and DoI.
> In addition, our method takes the whole computational graph into account, enabling us to include BERT’s components previously overlooked such as skip connections. Our methods are more generalizable to other models and tasks (with the definition of doi and qoi), without relying on specifics of model components as explanations, such as attention weights.
>
> 3) Other clarifications
>
> First, the difference between embedding and attention level graphs is that the former defines patterns over only the set of internal embeddings, while the latter adds nodes with all attention heads and skip connections. As a result, attention-level patterns abstract more of the model (by including one attention head or the skip connection in each layer while embedding level patterns essentially includes all attention heads and skip connections) thus drops more in accuracy. Various results in different tasks mainly prove our method is effective for a range of tasks, thus Table 1 is meant more as a comparison between different graph settings within each task, but we do understand the reviewer’s suggestion on comparing the results vertically to see why they vary across tasks. We will explore these questions in future work, and add more qualitative analysis for other tasks as in 4.2 in the final version.

---

> > ### Comment · AnonReviewer1 · 2020-11-24
> > **Response to authors**
> >
> > Thank you for your detailed responses and revisions -- things are substantially clearer now, and I think the contribution here is interesting. The preliminary analysis in Section 4.2 suggests exciting potential for insights into how these models handle different linguistic phenomena. Ideally I would like to see more space in the paper devoted to this type of analysis, since the couple of examples explored are intriguing, but don't give conclusive answers. But overall, the paper has improved substantially, and I am raising my score accordingly.

---

### Official Review · AnonReviewer4 · 2020-10-28
**Exciting work, I would appreciate a bit more extensive analysis and results section**

**Rating:** 6
**Confidence:** 3

**Review:**

*Summary*

This paper investigates how BERT uses attention to contextualise information. To find an answer to this question, the authors use abstractions of sets of paths through a neural network model, to quantify and localise the effect of specific inputs to the output. They describe these paths -- multi-partite patterns, as well as a guided pattern refinement that can be used to discover influential paths. To showcase their proposed techniques, the authors consider subject-verb agreement and reflexive anaphora in BERT.

*Review*

The proposed method is exciting: investigating the flow of information information in neural models is a promising research direction that is receiving more and more attention. The authors spend sufficient time explaining several aspects of their method, although I find it at times a bit confusing to understand why some information goes in the background section and some instead in the section "path abstraction".

In terms of the result section, I think the paper could be improved. It is not trivial to parse the numbers in Table 1 and 2, and not much explanation is given. The paper furthermore only reports results on one of the tasks given in the abstract, all the rest of the results are in the appendix. I am missing also some explicit comparisons with previous studies with similar aims (see appendix). In general, I find that a bit too much of information that is relevant to understand the story is put in the appendix.

*Questions for the authors:*
- Much work has been done on investigating how LSTM language models model subject verb agreement and anaphora resolution, and in the literature also  several comparisons between BERT and LSTM models can be found. To validate your technique, perhaps it could be interesting to run it also for LSTM models, for which it is easier to establish if the results make sense? _**Update after author response:** I understand that space might not permit running this also for an LSTM model, and that there are substantial differences between LSTM models and BERT, in terms of depth. Still, I would appreciate to have some baseline results, which would help to better assess the contribution. In my opion, LSTM models, for which much more is already known about the paths through the network, would be the easiest way to go._
- I think your work is very interesting, but I am really struggling to understand the results. Could you elaborate a bit more, and provide a bit more structured explanation and interpretation? _**Update after author response**: I appreciate that you took into consideration my request to add a bit more explanation. I still find the structure a bit difficult to understand, but it has definitely improved since the last version_

*A few notes and suggestions*
_**Update after author response:** Thanks for incorporating most of my suggestions_
- Figure 1 comes a bit early for me, at the point where it is referenced I cannot really parse it yet, because much of the info to understand follows only later
- The font size of the legend in Figure 1 is very small, could you increase it?
- You generate your own dataset with NA-tasks with fixed patterns, this was previously also done by Lakretz et al 2018 -- The emergence of number and syntax units in LSTM language models -- a dataset which was used by several others afterwards. For comparability with others that have investigated this phenomenon in BERT and other models, it might be nice to run some experiments using that dataset.
- A very relevant reference is Jumelet et al (2020) -- Analysing Neural Language Models: Contextual Decomposition Reveals Default Reasoning in Number and Gender Assignment -- who use contextual decomposition to find how information flows in a recurrent LM, looking also at SVA and RA. In a more recent paper Jumelet (2020) repeated this analysis also for BERT.  It would be nice to see some comparisons, and also some contextualisation on how the method here relates to their work, and to contextual decomposition (Murdoch et al 2018 -- Beyond Word Importance: Contextual Decomposition to Extract Interactions from LSTMs) in general.
- In related work, I am missing Hupkes et al 2018 (Visualisation and ‘diagnostic classifiers’ reveal how recurrent and recursive neural networks process hierarchical structure) as a reference for diagnostic classification
- bibliography should be checked, there are quite a few arxiv references that are actually published papers

_**Update after author response**: I think the paper has improved substantially. The direction is exciting, but even with the updates I believe this paper needs quite some work to improve the presentation and clarity, which is why I have not updated my score._

---

> ### Author Response · Authors · 2020-11-18
> **Response to Reviewer 4**
>
>  We appreciate all the suggestions and questions provided by the reviewer.
>
> Response to Reviews
> We agree with the reviewer that the first submission does not give an easy-to-follow introduction of the prior work and how it motivates the definition of the pattern influence and GPR. In the updated submission, we remove redundant introduction of BERT and make use of the extra space to integrate contents from the appendix. Specifically, we have 1) added a clear graph representation of BERT’s transformer layer (Fig 1 in new version), 2) introduce completeness axiom in detail in the Appendix B1 to let readers to understand convergence analysis means to show how many points are required to approximate the expectation with summation in practice, and 3) added more interpretations of Table 1 (we combined Table 1 and 2 from the last version into the Table 1 of the update) by clearing state the definition of concentration (Def. 5) and details to run model compression.
>
> Response to Questions
> 1)  The goal of our paper is to analyze the contextualization in BERT so we design our experiments and analysis specifically for BERT, leaving very few spaces to extend the discussion to other language models like LSTM. However, the key difference between Lu et. al 2020, which applies influence paths to LSTM language models and our work is that LSTM contains very few layers (1 or 2), therefore, it is easier to do an exhaustive search to locate the most significant path as is done by [Lu et. al 2020] so GPR may not be necessary.  However, we do agree it would also be interesting to show a head-to-head comparison on how BERT handles SVA differently from LSTM and why the former is superior in the relevant tasks. The relative experiments are running in the backlog and we aim to include in a future version.
>
> 2)In the updated submission, we have added more clarifications to help understand the results, the details of which can be found in the official comment.
>
>
>
> Response to notes and suggestions
>
> 1) & 2) We have moved figure 1 down in the updated version and increased the legend font.
>
> 3)  We agree with the reviewer’s suggestion about applying our method on other datasets. We source our dataset directly from Goldberg 2019, which evaluates BERTS against SVA and RA datasets under the MLM setting. We speculate our results and conclusions can be generalized to other similar datasets as well.  We have also added the reference to the related work section.
>
> 4) We have added some mentioned references to the related work section. Explanation NLP using contextual decomposition(CD) (Murdoch et al 2018) such as  Jumelet et al offers an alternative measure for explaining language tasks, especially interactions among words in a sentence,  in models such as LSTM. Our method exposes the internals of BERT using end-to-end gradient-based attribution methods while CD uses shapley values to linearize non-linear activation functions. The two methods have different explanation cores while we agree with the author’s suggestion on comparing the qualitative explanations of two methods.
>
> 5) We have added the reference to the related work section.
> 6) we have fixed the references and switched all corresponding proceedings

---

### Official Review · AnonReviewer2 · 2020-10-28
**Official Blind Review #2**

**Rating:** 6
**Confidence:** 3

**Review:**

### Summary:

This paper addresses the problem of how BERT contextualizes information. Specifically, this paper tackles a computational limitation of influence paths used to analyze LSTM models by introducing multi-partite patterns and guided pattern refinement (GPR). A multi-partite pattern, $\pi$, defines a set of paths that follow a sequence of edges. The more edges it has in the sequence, the fewer paths it defines. Then, GPR is a greedy algorithm that iteratively adds an edge to $\pi$ such that the influence of the pattern is maximized. For each iteration, the search is constrained to a set of edges within a single layer of BERT. As a result, an influence path is constructed (or rather approximated). The author validates the proposed method using concentration and model compression experiments for multiple agreement tasks. The concentration results show that influence paths discover by GPR have a very high influence. This is true for both embedding-level paths and attention-level paths. For model compression, the results show that most of the accuracy is maintained when retaining the identified influence paths and "churning" the others. Finally, the author explains how BERT contextualizes input words by visualizing the influence paths. The author discusses the four examples of the SVA across Obj task. The influence of the subject and the intervening noun is similar to that of LSTM (Lu et al., 2020). Insights from the visualization of the influence paths are discussed such as "copy and transfer" and a conflicting role of "that" (which might explain the common mistake of BERT).


### Recommendation:
Overall, I incline toward rejecting. This paper provides an instrument to explain BERT, but I have a hard time understanding the result itself (influence paths or patterns). I also have a major concern with the final analysis and its findings.

#### Pros:
- Novel method to compute influence paths that can help us interpret how BERT utilizes input words.
- The quantitative experiments conducted on many agreement tasks validate the proposed methods.
- Interesting findings that BERT likes to "copy".

#### Cons:
- Interpreting the influence paths to understand how BERT works is quite difficult and arguably subjective. It is also confusing why Figure 1 is so different from Figure 3.
- Although the result influence paths include the skip connections, it is not quite clear how it is compared to the attention approach in the other aspects. For example, the author could use the model compression experiment on the high attention weight embedding (or hidden representation).
- It is unclear how the main finding of section 4.2 came to be. For example, how many sentences that the author observed before concluding that "copy and transfer" are common. The speculation of "that" influence paths can somewhat explain the result discrepancies of SS and PS. Are there other words that behave similarly (such as "who")? I think the author should provide a more rigorous analysis on the interpretability of the proposed method, as well as an explanation of BERT contextualization.
- It is quite common to study how the model works through this kind of task. But, most of the time, we BERT of encoding sentences without [MASK] word. I am not sure how this finding will generalize to other tasks.
- This paper is very hard to read. The math expressions are clear, but the explanations and intuitions are often confusing.


### Questions:

1. I am not quite sure that "churning" means. Could you please provide more detail?
1. What are the numbers on the "Significant patterns" figures?
1. I am a bit confused with Figures 1 and 3. In Figure 1, there is a path from 1 word to [MASK], but in Figure 3 (and others), there are multiple paths. Could you explain why is there a difference?
1. Would you have the same result using Lu et al., average embedding instead of [MASK] embedding as a base embedding?

### Minor comments:
- I think Section 4.2 onward is a bit rushed. There are many typos, such as "... connections.  . We speculate ..." or "we how BERT".
- Definition 5, typo on "$n_w,n_w$"?
- So much useful figures and tables are in Appendices.
- I am not sure the point of the convergence analysis of the BERT base vs small.

---

> ### Author Response · Authors · 2020-11-18
> **Response to Reviewer 2**
>
> We appreciate all the suggestions and questions provided by the reviewer.
>
> Responses to Cons:
>
> Cons 1): We agree that subjectivity is hard to avoid in interpreting results from an explanation device.  Despite this concern, gradient-based methods provide an end-to-end and rigorous causal analysis from input to the output.  Influence paths (and patterns) is a step further by decomposing those gradient-based attribution methods to chains of internal components in BERT’s computation graph. We have since added more clarifications in the background and method sections to show how influence paths help us understand contextualization in BERT. In the original submitted version, we visualize the influence patterns for individual sentences in Fig 1 and an aggregated visualization for patterns across sampled sentences in  Fig 3. We realize the aggregated visualization might be confusing to readers, Therefore, in the updated version we only retain Fig 1 (a.k.a. Fig 2 in the most updated version) to support our analysis and have moved all aggregated visualizations to the appendix.
>
> Cons 2): We have done experiments creating baseline patterns with high attention weights in each layer and discovered the derived patterns are inconsistent and random across examples, also far worse compared to the influence-based patterns. Patterns derived from attention weights would also be less meaningful since the attention blocks across layers are disconnected (separated by a dense layer). We will include a more rigorous comparison to the final version given more space.
>
> Cons 3): Both individual and aggregated visualizations(over 200 samples) show a lot of dashed horizontal lines (copy) and solid lines only at the last few layers (transfer), which give us a qualitative view of the overall trend.  The behavior of ``that’’ in object relative clauses is one example of how influence patterns can be used to understand contextualization for SVA, while we will include more examples as such in the final version given more space.
>
> Cons 4): The analysis of BERT is studied in both the MLM paradigm (predicting the [MASK]) and also the representation learning paradigm (without  [MASK]). Our pattern abstraction can be applied to the latter as long as we can formulate a quantity of interest (QoI) with respect to an output of the model. Since there are no probabilistic outputs of the latter paradigm to define a meaningful qoi like s(is) - s(are), we can potentially compute qoi over the output of the probing classifiers trained on the representations (such as the ones in Tenny 2019a associating internal representations of BERT with various grammatical concepts like POS and NER tagging).
>
> Cons 5): We agree with the reviewer that our motivations and results can be structured in a better way. Please feel free to take a look at our updated submission.
>
>
> Responses to Questions:
>
> 1) We have replaced “churning” with “replacing” in the updated version. Essentially for compression, we replaced the nodes outside the significant pattern with [MASK] or zero vectors.
>
> 2) They refer to the attention head number the pattern influence flows through.
>
> 3) The difference between Figure 1 and 3 in the originally submitted version is explained above.
>
> 4) We speculate that Lu et al. 2020 chooses the average embedding because of the absence of “meaningless” token like [MASK] in RNN language models, and since average embedding is equidistant from both singular and plural nouns, they represent a “number neural state”. However, [MASK] in the MLM setting of BERT is a better choice for representing a ``blank state baseline,  similar to a black image for vision tasks (a widely used baseline in explaining vision models).  Another reason for choosing [MASK] over the averaged embedding is that we also computed influence for all other words besides nouns, and words like "`that" are harder to find an average embedding comparable to numbered nouns. We have actually conducted experiments with the averaged embedding as the base embedding and will include the comparison in the final versions’ appendix.
>
> Responses to Minor concerns:
>
> We thank the reviewer for pointing out the typos and we will select more figures and illustrations from the appendix in the final version. The convergence analysis in the appendix demonstrates how well the summation in Def 1 approximates the integral when using DoI as a uniform distribution over a linear path. The conclusion is that using summation to approximate the integral is not easily achievable in $BERT_{base}$ with sample size < 10k but much more accurate in $BERT_{small}$ with around 50-100 points (shown in Fig. 6).  We have clarified this convergence analysis in the updated version.

---

> > ### Comment · AnonReviewer2 · 2020-11-23
> > **Response to author's rebuttal**
> >
> >  Although it was odd that the paper changed substantially from the original submission, I appreciated the updated introduction and background sections. The response to other reviewers and my review on Section 4.2 was a bit vague. I was hoping for a more quantitative affirmation of the suggested findings. Overall, the author clarified many concerns, especially, the confusing aspects of the influence path visualization and the choice of the base embedding. I'll increase my score accordingly.

---

### Author Response · Authors · 2020-11-18
**Official Comment to All Reviewers**

To all reviewers:

We greatly appreciate the feedback from all reviewers. We have updated our submission to resolve concerns raised by reviewers and we would like to use this channel to 1) answer the common concerns across different reviewers, 2) and summarize the difference compared with our last submission.

Responses to Common Concerns:
1) One concern is the clarification of results and how they show BERT’s contextualization of various linguistics concepts. We have since made a clear link between the key questions, results and takeaways.
2) We will make head to head comparisons to baselines such as RNN models or attention-based explanations in the final version.
3) We have fixed typos, confusions and other stylistic issues mentioned by the reviewers.

Summary of Update:
1) Introduction:
We have introduced central questions and define the purpose of relevant metrics for verifying our abstractions.
2) Background and Method:
We have provided more background to attribution methods and justifications for our choice of input influence parameters. We have clarified our GPR algorithm with more systemic notations.
2) Evaluation:
We have added more details and justifications for the datasets and parameters used, as well as more clarifications for the figure. We have not added any new results in this version, but we will leverage the reviewer’s comments to add more qualitative results and baseline comparisons given more space in the final version.

---

### Decision · Program_Chairs · 2021-01-07
**Final Decision**

**Decision:**

Reject

**Comment:**

This paper propose a method to explain the contextualization of BERT by identifying a set of influence paths from the input to the output.  Although all reviewers give overall score 6, their comments are pointing to the negative direction.  The following excerpts summarize the general sentiment of the reviews:
R2: Overall, I incline toward rejecting. This paper provides an instrument to explain BERT, but I have a hard time understanding the result itself (influence paths or patterns). I also have a major concern with the final analysis and its findings.
R4: I think the paper has improved substantially. The direction is exciting, but even with the updates I believe this paper needs quite some work to improve the presentation and clarity, which is why I have not updated my score.
R1: Overall, I think there are certainly interesting analyses that could come out of this work, but the current paper does not provide a clear enough contribution to be ready for publication.
R3: It looks like the proposed method is largely based on (Lu et al, 2020); the major difference is the introduction of Multi-partite patterns, which basically expands the objects of influence analysis from paths to patterns (partial paths). This doesn't look like a significant novelty.